# Modelling the climatic suitability of Chagas disease vectors on a global scale

**Fanny E Eberhard[1,2]\*, Sarah Cunze[1,2], Judith Kochmann[1,2], Sven Klimpel[1,2]**

[1]Goethe University, Institute for Ecology, Evolution and Diversity, Frankfurt, Germany; [2]Senckenberg Biodiversity and Climate Research Centre, Senckenberg Gesellschaft für Naturforschung, Frankfurt, Germany

**Abstract** The Triatominae are vectors for *Trypanosoma cruzi*, the aetiological agent of the neglected tropical Chagas disease. Their distribution stretches across Latin America, with some species occurring outside of the Americas. In particular, the cosmopolitan vector, *Triatoma rubrofasciata*, has already been detected in many Asian and African countries. We applied an ensemble forecasting niche modelling approach to project the climatic suitability of 11 triatomine species under current climate conditions on a global scale. Our results revealed potential hotspots of triatomine species diversity in tropical and subtropical regions between 21°N and 24°S latitude. We also determined the climatic suitability of two temperate species (*T. infestans*, *T. sordida*) in Europe, western Australia and New Zealand. *Triatoma rubrofasciata* has been projected to find climatically suitable conditions in large parts of coastal areas throughout Latin America, Africa and Southeast Asia, emphasising the importance of an international vector surveillance program in these regions.

## Introduction

The Triatominae are haematophagous insects of the order Hemiptera and comprise 152 species subdivided into 16 genera, including two fossils (*Poinar, 2013*; *Mendonça et al., 2016*; *da Rosa et al., 2017*; *Monteiro et al., 2018*). Triatomines are mainly distributed in Central and South America inhabiting environments ranging from tropical to temperate regions with cold winters (*de la Vega and Schilman, 2018*). Eleven species have also been recorded in the southern United States (*Curtis-Robles et al., 2018*). In the Old World, species belonging to the genus *Linshcosteus* occur in India, whereas eight species of the genus *Triatoma*, especially the cosmopolitan *T. rubrofasciata* occur in Africa, the Middle-East, Southeast Asia, and in the Western Pacific (*Gorla et al., 1997*; *Monteiro et al., 2018*).

Through their haematophagous lifestyle, triatomines function as vectors for pathogens such as *Trypanosoma conorhini*, *T. rangeli* and *T. cruzi*, the aetiological agent of Chagas disease (American trypanosomiasis) (*Deane and Deane, 1961*; *Ferreira et al., 2015*; *Vieira et al., 2018*). The flagellated protozoan parasite *T. cruzi* is transmitted by infectious faeces of the triatomines, which are rubbed into the bite wound. Further transmission routes include oral infection by the consumption of contaminated food or raw meat of infected mammalian hosts, congenital infection and transmission through blood transfusion or organ transplantation. Acute symptoms include fever, fatigue, headache, and myalgia, with long-term effects such as acute and chronic chagasic heart disease and gastrointestinal manifestations being more severe (*Nóbrega et al., 2009*; *Coura and Viñas, 2010*; *Carlier et al., 2011*; *Rosas et al., 2012*). *Lee et al., 2013* calculated a global economic loss of $7.19 billion per year attributable to Chagas disease due to high health-care costs and lost productivity from early mortality. It is estimated that 6 to 7 million people worldwide are infected with Chagas disease, most of them living in Latin America. Caused by global immigration and travel, Chagas disease has been increasingly detected in the United States, Canada, European and some Western

\*For correspondence:
Eberhard@bio.uni-frankfurt.de

**Competing interests:** The authors declare that no competing interests exist.

Pacific countries (*WHO, 2019*). However, due to the lack of vectors, there has been no vector-borne transmission outside of the Americas. This could change if the spread of the disease is followed by a propagation of the vectors. Although the mobility of triatomine bugs is generally limited, they can be passively transported by the shipment of infested goods and animals or luggage and along air transportation routes (*Fleming-Moran, 1992*; *Coura and Viñas, 2010*; *Pinto Dias, 2013*).

Among triatomine species, *Triatoma rubrofasciata* is most widely distributed and recorded from the United States of America, Central and South America, coastal regions of Africa and the Middle-East, the Atlantic Ocean (Azores) and port areas of the Indo-Pacific region (*Dujardin et al., 2015a*). It is the only member of the *Triatoma* genus occurring in the New and Old World, with frequent records in previously non-endemic areas (*Liu et al., 2017*; *Huang et al., 2018*). The evolutionary origin of *T. rubrofasciata* is unclear, although a New World origin seems more likely (*Dujardin et al., 2015b*). It is believed that through its close association with domestic rats (especially *Rattus rattus*), *T. rubrofasciata* was spread along the shipping routes of the 16th to 19th century (*Schofield, 1988*; *Gorla et al., 1997*; *Patterson et al., 2001*). *T. rubrofasciata* is able to transmit *Trypanosoma cruzi* in Latin America; however, there are no records of vector transmission in the Old World (*Dujardin et al., 2015b*).

In the past, Chagas disease vectors have frequently experienced range expansions (*Pinto Dias, 2013*). For instance, *Rhodnius prolixus*, one of the most important vectors of Chagas disease, was carried from Venezuela to Mexico and Central America by sea commerce and possibly by bird migration (*Hashimoto and Schofield, 2012*). In the light of ongoing globalization, changing climate and a concomitant shift of trade and travel routes, the probability of further migration of triatomine bugs and especially *Triatoma rubrofasciata* increases. This entails the risk of vector-associated transmission of Chagas disease (*Schofield et al., 2009*). Therefore, constant monitoring of the vectors, but also the determination of potentially suitable habitats for these vectors, appears to be of great importance. The potential spread of various triatomine species under current and future climatic conditions in South, Central and North America has been extensively studied, but there is a lack of knowledge about climatic suitability outside the Americas (*Gurgel-Gonçalves et al., 2012*; *Garza et al., 2014*; *Parra-Henao et al., 2016*). The aim of this study was to investigate the climatic suitability under current climatic conditions for eleven triatomine species on a global scale using ecological niche modelling (ENM). We concentrated on domestic and peri-domestic triatomine species representing different biogeographical regions and deemed to be the main vectors of Chagas disease. An ensemble forecasting approach was applied (*Araújo and New, 2007*), which is considered to yield robust estimations of the habitat suitability. In this way, we are able to identify areas at risk, pinpoint triatomine species which find suitable habitats outside their current range and therefore might possess a high potential for expansion.

## Results

### Potential distribution under current climate conditions

Global species distribution modelling revealed several regions with current suitable climatic conditions for the considered triatomine species. Comparing the modelled potential distribution of the species, differences in the preference of climatic conditions are evident. *Rhodnius brethesi*, *R. ecuadoriensis* and *Triatoma maculata* are limited to one or a few areas with mostly tropical climate. *Triatoma brasiliensis*, *Panstrongylus geniculatus*, *P. megistus*, *R. prolixus*, *T. dimidiata* and *T. rubrofasciata* find suitable climate conditions in a broad range of tropical and sub-tropical regions, while *T. sordida* and *T. infestans* possess a broad potential range in temperate regions (*Figure 1*).

The projected range of *R. brethesi* and *R. ecuadoriensis* is limited to areas with equatorial, tropical wet climate. In the case of *R. brethesi*, these areas include, above all, the Amazon region in South America and Southeast Asia (Indonesia, Malaysia, New Guinea) (*Figure 1A*). *R. ecuadoriensis* shows only a small modelled potential distribution in western Ecuador, parts of Indonesia, Malaysia and Papua-New Guinea as well as the Congo Basin (*Figure 1B*). Beyond its observed distribution area in Venezuela, Guyana, Suriname and French Guiana, *T. maculata* possesses distribution potential under current climate conditions in the North of Brazil, Peru, Central Africa and parts of Southeast Asia (Malaysia, Indonesia, Philippines, New Guinea) (*Figure 1C*). *T. brasiliensis* prefers dry and wet savannah climate as found in eastern Brazil. Furthermore, this species has a modelled climatic distribution

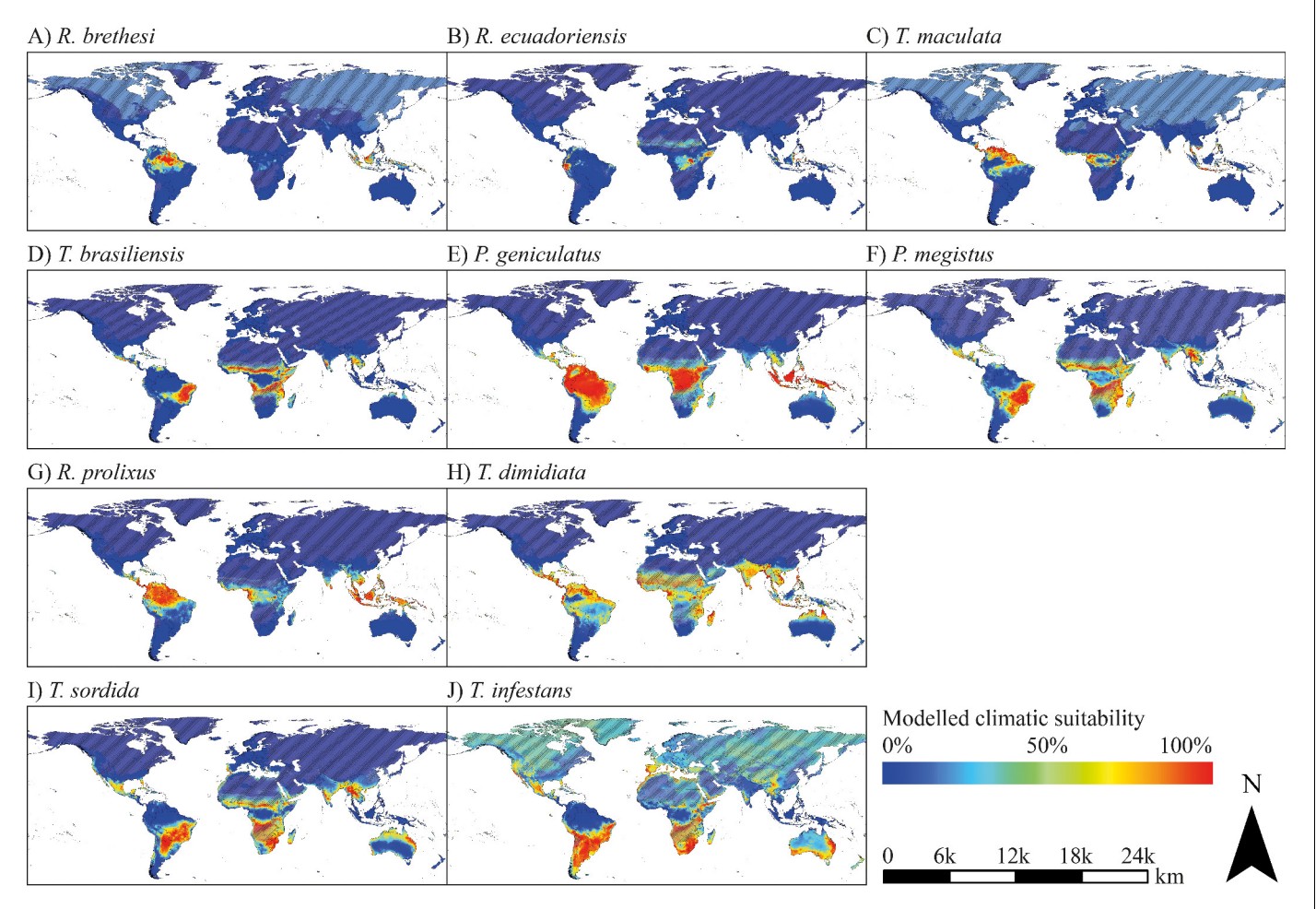

**Figure 1.** Modelled current climatic suitability. (A–J) Modelled climatic suitability (consensus map) of 10 triatomine species under current climate conditions. Hatched areas indicate regions where the projection is uncertain. Maps were built using WGS 84 as geographical system and ESRI ArcGIS (**ESRI, 2018**).

potential in southern West Africa, northern and southern Central Africa and East Africa (*Figure 1D*). Models project large climatically suitable areas for *P. geniculatus*, *P. megistus*, *R. prolixus* and *T. dimidiata* in tropical regions in South America, Central America, Caribbean, Central and East Africa, eastern Madagascar, in the South of India and Sri Lanka and throughout Southeast Asia (*Figure 1E–H*). In addition to species preferring tropical, wet climate, species that occur in temperate, semi-arid regions have also been modelled. These include *T. sordida* and especially *T. infestans*. The climatic suitability models of the latter show extensive potential spread in both semi-arid to humid and temperate to cold regions comprising the Southern Cone of South America, parts of Brazil, Bolivia and Peru, Mexico, Caribbean and Florida (USA), the South of Africa, parts of the Arabian Peninsula, the West and South of Australia, New Zealand and in Europe Portugal, Spain, France, Italy, Greece, Ireland and Great Britain (*Figure 1J*). For *T. rubrofasciata*, primarily coastal regions in tropical savannah and monsoon areas were predicted as climatically suitable including the east coast of South America and Central America, large parts of Brazil and Venezuela, Caribbean, Florida (USA), the coasts of Central and Eastern Africa, the eastern coast of Madagascar, southern India and Sri Lanka, Thailand, Malaysia and Indonesia, Vietnam, southern China and Japan, Philippines and the eastern coast of Australia (*Figure 2*).

The observed occurrence of the considered species is mainly consistent with the projected climatic suitability in Latin America. Nevertheless, it is noteworthy that for some species the modelled climate suitability in Central and South America exceeds the area of current occurrence. For

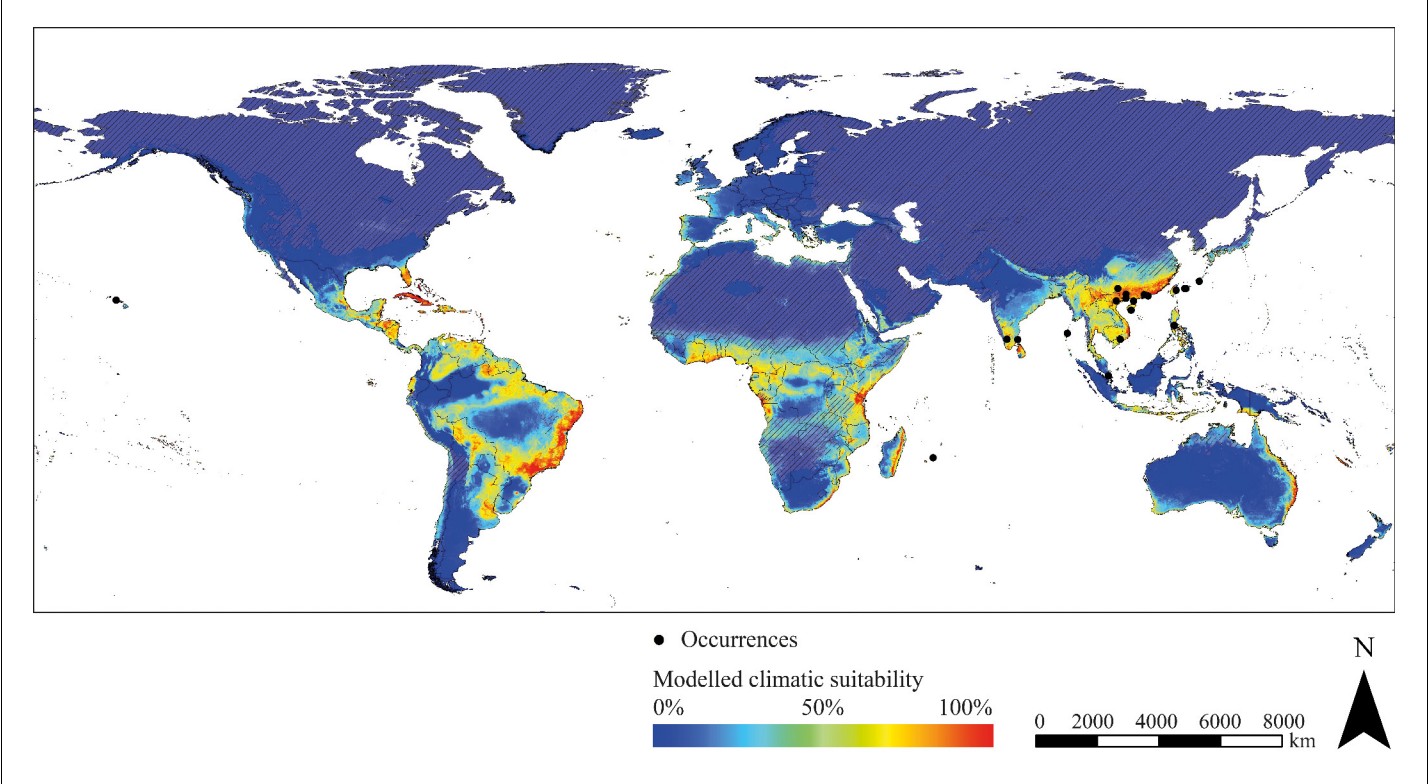

**Figure 2.** Modelled current climatic suitability of *T. rubrofasciata* (consensus map) and observed occurrence records outside the Americas. Hatched areas indicate regions where the projection is uncertain. Maps were built using WGS 84 as geographical system and ESRI ArcGIS (*ESRI, 2018*).

example, *T. dimidiata* is not observed in large parts of Brazil, Peru and Bolivia, although a good climatic suitability has been projected (*Figure 1H*).

Potential hotspots of triatomine diversity are revealed by the species diversity map displaying the number of the modelled triatomine species that find suitable climatic conditions in the respective regions. The quantity of species varies between zero and seven. Areas of great triatomine diversity have foremost tropical forest and savannah-like climate and include predominantly regions between 21°N and 24°S latitude (*Figure 3*). However, even in temperate regions, some species could find suitable climatic conditions, for example in Portugal, Spain and eastern Australia.

The most important bioclimatic variable is the temperature seasonality (BIO4) for all considered species, closely followed by the minimum temperature of the coldest month (BIO6) and maximum temperature of the warmest month (BIO5). The three precipitation variables (BIO13, BIO14, and BIO15) seem to shape the species distribution in a subordinate way.

## Model evaluation

The evaluation of the global projection of the climatic suitability shows that almost all actual occurrence points of *T. rubrofasciata* (with coordinates provided) are within an area classified as climatically suitable by the models (*Supplementary file 1*). This is particularly evident in South India, Vietnam, South China, Taiwan or the Philippines. A few occurrence points are located in areas projected as less suitable including seaports such as Singapore and the Okinawa islands in Japan. According to the models, every country in which *T. rubrofasciata* has been found (but without specific coordinates of occurrence records given) provides at least one area offering suitable climatic conditions, for example Indonesia, Madagascar and several African countries.

The discriminatory capacity of all models displays a good predictive performance, which is also reflected in the AUC values of over 0.9. However, differences in the performance of the models become apparent. GBM (generalised boosted models) performs particularly well for all species and

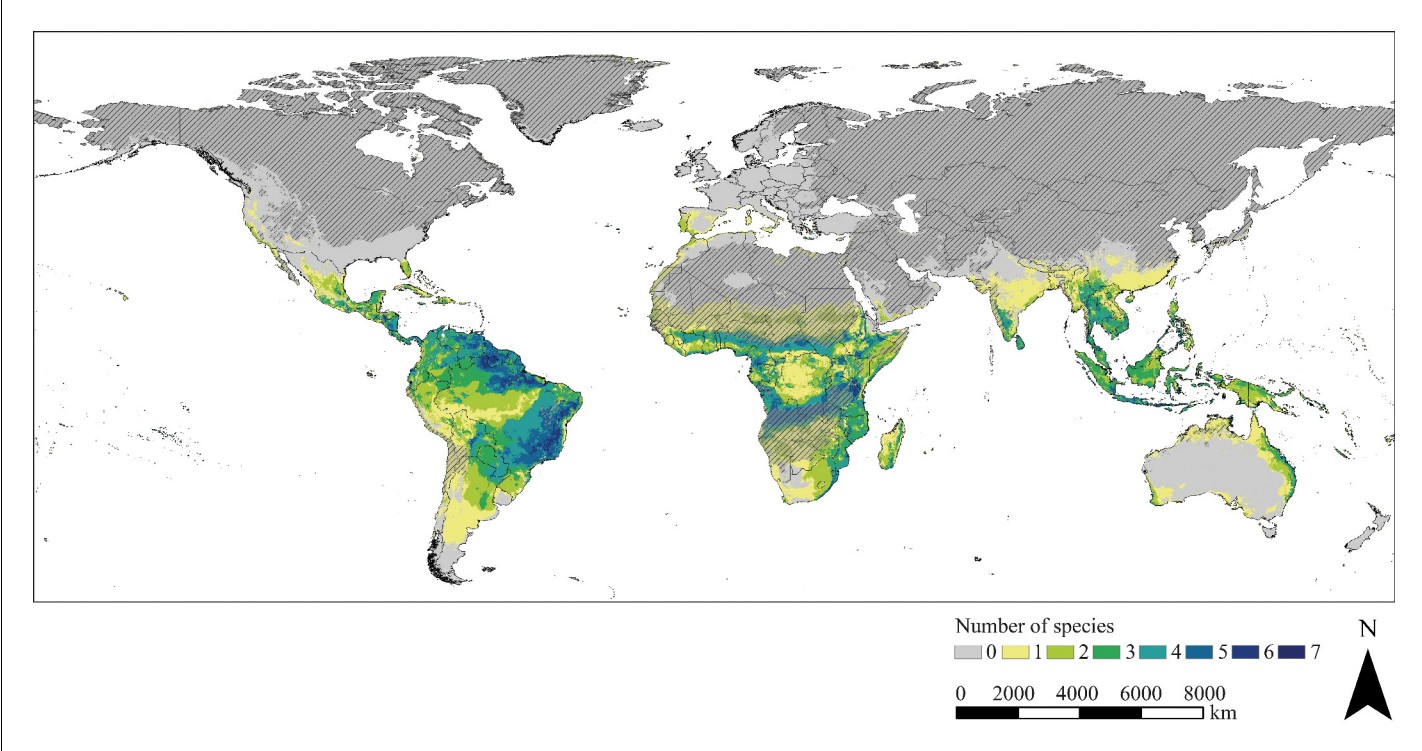

**Figure 3.** Species diversity. The map is based on the combined binary modelling results highlighting potential hotspots of triatomine species diversity. Hatched areas indicate regions where the projection is uncertain. Maps were built using WGS 84 as geographical system and ESRI ArcGIS (***ESRI, 2018***).

achieves the highest AUC values – the same applies to GAM (generalized additive models). Slightly lower AUC values are generated by ANN (artificial neuronal networks) and MAXENT (maximum entropy) (***Supplementary file 2***, ***Supplementary file 3***).

## Discussion

With a few exceptions, Triatominae are currently widespread in Central and South America where they transmit the causative agent of Chagas disease, *Trypanosoma cruzi*. However, it is not yet known whether areas outside the Americas provide suitable habitats for triatomine species. This study analyses the potential geographical distribution of 11 triatomine species under current climatic conditions on a global scale. For this purpose, we used ENM to identify regions that offer climatically suitable conditions for the examined species.

Despite climatic requirements, the triatomine species have different microhabitat preferences and host spectra and therefore, possess a dissimilar vector potential. For instance, *R. brethesi* is closely associated with palm trees and features prominently in the sylvatic transmission of *Trypanosoma cruzi* feeding particularly on opossums (*Didelphis* spp.). Domestic or peri-domestic behaviour is not observed, thus, *R. brethesi's* domestic vector potential is most likely low (***Coura et al., 2002***; ***Rocha et al., 2004***). *Rhodnius ecuadoriensis* is distributed in southern Ecuador and northern Peru, where it exhibits domestic and peri-domestic behaviour invading chicken coops and human dwellings. It establishes dense populations and is commonly infected with *T. cruzi* indicating a high vector potential. In the sylvatic environment, *R. ecuadoriensis* is mainly found in palm trees (*Phytelephas aequatorialis*) (***Abad-Franch et al., 2005***). The *T. cruzi* infection rate of *Triatoma maculata* depends on the geographical area. In most regions *T. maculata* has ornithophilic feeding preferences, whereas studies in the Colombian Caribbean region reported active transmission in peri-domestic human dwellings involving dogs as reservoir hosts (***Cantillo-Barraza et al., 2014***; ***Cantillo-Barraza et al., 2015***). For these three species, only a few regions outside the Americas with distinct tropical climate have been projected as climatically suitable including the Congo Basin in Central

Africa and a few parts of Southeast Asia. Here, *T. maculata* comparatively shows the largest distribution comprising potentially suitable areas within tropical rainforest and savannah climates (*Figure 1C*).

*Triatoma brasiliensis* is located in tropical savannah regions with a hot and dry climate altering with intensive rain. There, it is found under rock piles feeding on a broad range of reptile and mammalian hosts, including humans. Due to its high *Trypanosoma cruzi* infection and intradomiciliary infestation rates, *Triatoma brasiliensis* is considered as an important vector for Chagas disease (*Carcavallo et al., 1997*). The climatic niche of *Panstrongylus geniculatus* was projected to be very broad (*Figure 1E*). A result that is reinforced by the literature in which the species is described as eurythermic and adapted to several dry as well as humid ecotopes (*Patterson et al., 2009*). It feeds on the blood of various hosts, such as marsupials, opossums, anteaters, armadillos, bats, cats, birds and chicken, in whose nests and coops it can be found. Additionally, it invades human houses and possesses a high susceptibility to *Trypanosoma cruzi* (*Feliciangeli et al., 2004*). Similar climatic distribution patterns are shown by the anthropophilic species *R. prolixus*, which permanently colonises human dwellings (*Rabinovich et al., 2011*). *P. megistus* occurs primarily in the Atlantic rain forests of South America and requires high relative humidity for breeding. With exception of the triatomine species occurring in temperate climate regions, solely the modelled distribution of *P. megistus* is highly influenced by the bioclimatic variable describing the maximum temperature of the warmest month (BIO5). The reason could be that the species is strongly limited by arid climates (*Forattini, 1980*). Outside of South and Central America, the models of these six species show widespread potential distribution areas with suitable climate conditions in the Caribbean, Central and East Africa, Madagascar, South India and throughout Southeast Asia.

*Triatoma infestans* and *T. sordida* find climatically suitable habitats in subtropical, but also temperate regions as they exist in the southern cone of South America and northern Central America, respectively. They are also the only triatomine species for which habitats in southern Europe, eastern Australia and large parts of South Africa have been projected to be climatically suitable (*Figure 1I–J*). *T. infestans* is adapted to deal with cold temperatures and therefore, has a greater diversity of wild habitats, which is probably related to the behavioural plasticity of the species. It shows exceptional domestic behaviour and is classified as one of the most important Chagas disease vectors in South America (*Brenière et al., 2017*; *Belliard et al., 2019*). *T. sordida* also has a propensity for peri-domestic behaviour in the absence of primary domestic vector species, thus, it often occurs when *T. infestans* is eradicated from human dwellings (*Diotaiuti et al., 1993*; *Galvão and Justi, 2015*).

Previous niche modelling approaches focused mainly on the potential distribution in North, Central and South America neglecting the actual global distribution achieved by *T. rubrofasciata* and the risks emerging from other Triatominae. For example, *Garza et al., 2014* predicted a potential northern shift in the distribution of *T. gerstaeckeri* and a northern and southern distributional shift of *T. sanguisuga*, both important vectors of *T. cruzi* in the United States, under future climate conditions. Similar studies have been conducted for regions in Brazil, Mexico, Colombia, Chile and Venezuela in Central and South America (*Sandoval-Ruiz et al., 2008*; *Gurgel-Gonçalves et al., 2012*; *Hernández et al., 2013*; *Ceccarelli and Rabinovich, 2015*; *Parra-Henao et al., 2016*).

In general, the projected climatic habitat suitability reflects the realised species distribution in South America exceptionally well (*Supplementary file 4*). In some areas, however, the models appear to slightly overestimate the potential distribution as it could be noted in the modelling of *T. dimidiata*. Although eastern and western South America were projected as climatically suitable, the observed distribution of *T. dimidiata* covers exclusively Central America and northern South America. Such discrepancies could be ascribed to factors not related to climate conditions, such as dispersal limitations and interspecific competition. Furthermore, highly effective vector control measurements such as indoor residual spraying and housing improvements significantly reduce the distribution of triatomine insects in areas that would otherwise be considered a suitable environment (*Cucunubá et al., 2018*). Sampling biases, such as insufficient or inhomogeneous sampling can result in species distribution modelling reflecting sampling effort rather than actual distribution. Nevertheless, the occurrence data used were taken from a published atlas of the Chagas disease vectors and, thus, represent a comprehensive and reliable source (*Carcavallo et al., 1998*; *Fergnani et al., 2013*). The applied ensemble forecast approach is considered to return more robust estimations compared to single algorithms and is therefore an eligible tool for ENM. Nevertheless, like all

modelling approaches, ensemble forecasting is also subject to certain restrictions. This includes in particular the generation of pseudo-absences, since reliable absence data were not available. We have sought to minimize this problem by using a geographically filtered absence selection, where pseudo-absences are sampled in a defined spatial distance to occurrence points. The fact that explanatory variables are only available up to the year 2000 entails that the modelled distribution could shift slightly under present conditions. Thus, the distribution of thermophilic species might be underestimated due to a changing climate. Areas in which the projection is uncertain are additionally displayed. It is striking that this affects particularly dry and cold regions, such as the Atacama Desert, the Sahara, the Namib and Kalahari Desert, Antarctica, or parts of the Arabian Peninsula. The reason for this is probably the extreme maximum and minimum temperatures as well as the precipitation patterns in these areas, which deviate significantly from the values of the bioclimatic variables used for model training. The comparison between the global projected climatic suitability and the actual occurrences of *T. rubrofasciata* imply that the algorithms also project well outside the Americas. Almost all occurrence records are located in areas projected to be at least partially climatically suitable (*Figure 2*). Occurrence points located in areas with less projected suitability often occur in large trade centres, such as Singapore where individual triatomines could be introduced by shipping or air transport. This could be also true for smaller islands, such as the Japanese Okinawa islands. Whether established populations exist there is not affirmed. Each of the countries where *T. rubrofasciata* has been reported but no coordinates are available possess at least one area projected as climatically suitable for *T. rubrofasciata*. This applies especially to coastal regions of Angola, Cambodia, China, Comoros, the Republic of the Congo, the Democratic Republic of the Congo, Guinea, India, Indonesia, Japan, Madagascar, Malaysia, Martinique (France), Mauritius, Myanmar, Philippines, Saudi Arabia, Seychelles, Sierra Leone, Singapore, South Africa, Sri Lanka, Taiwan (China), Tanzania, Thailand, Tonga and Vietnam. Nevertheless, the occurrence data for *T. rubrofasciata* is very scarce and probably incomplete due to a lack of monitoring. Therefore, occurrence points from 1958 to 2018 were used for validation including data collected outside the period in which the climatic variables were recorded. This could lead to a slight shift between the actually observed occurrence of *T. rubrofasciata* and the projected climatically suitable habitat. For example, it is possible that environments that were modelled as climatically unsuitable for the period 1970 to 2000 now have climatic conditions suitable for *T. rubrofasciata*.

The diversity map, which was compiled based on the binary modelling results, highlights regions possessing suitable climatic conditions for various triatomine species in tropical regions especially between 21°N and 24°S latitude. In South America, diversity hotspots are for the most part modelled for regions south of the equator. Indeed, a direct correlation exists between temperature and triatomine species richness leading to a significant increase of diversity from the poles towards the equator. This effect seems to be pronounced in the southern hemisphere of South America (*Rodriguero and Gorla, 2004*). According to *Péneau et al., 2016* highly diverse vector communities as well as less diverse communities can lead to peaks of Chagas transmission, while intermediate levels of triatomine diversity lowers the risk of transmission. This correlation is mostly associated with a dominant, highly vector competent key species being more abundant in less diverse communities. An increase in biodiversity reduces the contribution of this key species to the Chagas disease transmission rate, while the contribution of secondary species increases. At intermediate levels of biodiversity, this does not compensate the reduced risk associated with the key species, whereas in highly diverse communities, the contribution of the secondary species to the transmission rate nearly matches the contribution of the key species. Outside their native range, highest triatomine species richness can also be found in tropical regions. In Africa, the regions of greatest triatomine diversity are projected north and south of the equator in a tropical savannah climate similar to the climate south of the equator in South America (*Geiger, 1961*). This climatic characteristic also applies to Southeast Asia, where diversity hotspots are less prominent. Furthermore, our findings indicate that there are suitable climatic habitats for triatomine species in temperate areas, such as Portugal, Spain and Italy in Europe or eastern Australia.

The results of the relative contribution of the bioclimatic variables correspond to findings of other studies, which also identify temperature seasonality (BIO4) as an important determinant of triatomine species distribution. This is probably attributable to physiological limiting factors, such as the temperature-dependent development and temperature-induced dispersal stimulation of the triatomines (*Diniz-Filho et al., 2013*; *Pereira et al., 2013*; *Ceccarelli et al., 2015*). Whether temperature or

precipitation have a higher impact on triatomine distribution appears to be dependent on the considered species and further ecological factors (*Gorla, 2002*; *Bustamante et al., 2007*; *de la Vega et al., 2015*). For instance, although its lifecycle is not bound to water unlike many insects, a high relative humidity seems to be crucial for the development of *T. vitticeps*, a triatomine species occurring in the Atlantic forests of Brazil (*de Souza et al., 2010*). Our results indicate a low influence of the precipitation variables (BIO13-15) on the triatomine distribution. However, species interactions and other than the climatic influences on the Triatominae are not taken into account with this approach. For example, the niche of host species as well as microclimatic effects and habitat structures play an important role in the distribution of triatomine species.

Through our work, the global climatic suitability for many triatomine species has been demonstrated by ENM. Based on the results of this study, the ability to transmit *Trypanosoma cruzi* and the wide distribution achieved, *Triatoma rubrofasciata* currently seems to be the potentially most perilous source of autochthonous Chagas infections outside of the Americas. However, it has not yet been possible to provide evidence of such disease transmission (*Rebêlo et al., 1998*; *Dujardin et al., 2015b*). Due to their limited dispersal by flight and the fact that they do not have resting stages, transport of triatomine bugs seems to be very rare. Nonetheless, older instars of some triatomine species are able to survive several months without a blood meal and therefore, might survive longer transports (*Costa and Perondini, 1973*; *Cortéz and Gonçalves, 1998*; *Almeida et al., 2003*). Vulnerable dispersal routes could be shipping traffic fostering the distribution of coastal triatomine species as shown for *T. rubrofasciata*, but also trade and travel by air traffic. Our results, as well as the immigration of infected people, show that the potential transmission of *Trypanosoma cruzi* is not necessarily a sole Latin American problem (*WHO, 2019*). It has been estimated that there are >80,000 individuals infected with Chagas in Europe and the Western Pacific region, >300,000 in the United States, >3000 in Japan and >1500 in Australia (*Coura and Viñas, 2010*). A further spread of *Triatoma rubrofasciata* into regions with suitable climatic conditions, but also the possible introduction of other Chagas vectors in non-endemic areas, could aggravate the situation and increase the number of infections. In particular, regions offering suitable climatic conditions to a large number of different triatomine species are at risk. Therefore, it may be beneficial to establish national and international vector surveillance programs to monitor the spread of vectors, in particular for *T. rubrofasciata* as it is implemented in southern China (*Liu et al., 2017*), and to register Chagas disease as a reportable disease.

# Materials and methods

### Key resources table

| Reagent type (species) or resource | Designation | Source or reference | Identifiers | Additional information |
|---|---|---|---|---|
| Software, algorithm | RStudio | *R Development Core Team, 2019* | RRID:SCR_000432 | |
| Software, algorithm | ArcGIS for Desktop | *ESRI, 2018* | RRID:SCR_011081 | |
| Software, algorithm | biomod2 package | *Thuiller et al., 2019* | | Available at https://cran.r-project.org/package=biomod2 |

### Occurrence data

Eleven triatomine species were considered for species distribution modelling representing different biogeographical regions in Latin America. The selection was also based on their importance as Chagas disease vectors and their presence in domestic and peri-domestic environments.

Occurrence data of the triatomine species were obtained from data provided by *Fergnani et al., 2013*. This American distribution dataset contains point data for each species with associated coordinates and was generated to study patterns on morphological diversity and species assemblages in Neotropical Triatominae (*Fergnani et al., 2013*). In total, 4155 unique occurrence points were provided ranging from 31 for *Rhodnius ecuadoriensis* to 1180 for *Panstrongylus geniculatus* (*Table 1*, *Supplementary file 5*). *Fergnani et al., 2013* abstracted the occurrence data from distribution data from the 'Atlas of Chagas disease vectors in the Americas' (*Carcavallo et al., 1998*). In this atlas, the distribution of the species in the Americas is presented as detailed maps. These maps were copied

**Table 1.** Model specifications.
Occurrence points for all considered species used for modelling and model evaluation (AUC).

| Species | Occurrence records | AUC ensemble models |
|---|---|---|
| *Panstrongylus geniculatus* | 1180 | 0.985 |
| *Panstrongylus megistus* | 401 | 0.976 |
| *Rhodnius brethesi* | 85 | 0.991 |
| *Rhodnius ecuadoriensis* | 31 | 0.989 |
| *Rhodnius prolixus* | 540 | 0.981 |
| *Triatoma brasiliensis* | 178 | 0.994 |
| *Triatoma dimidiata* | 300 | 0.962 |
| *Triatoma infestans* | 631 | 0.977 |
| *Triatoma maculata* | 132 | 0.992 |
| *Triatoma rubrofasciata* | 268 | 0.98 |
| *Triatoma sordida* | 409 | 0.978 |
| Total | 4155 | |

and digitised at a 0.1° x 0.1° resolution and converted into a grid comprising the information of occurrence for each grid cell using an equal area Mollweide map projection. With the help of the map projection, occurrence points with coordinates were created (*Fergnani et al., 2013*).

In order to assess the reliability and completeness of the data obtained from *Carcavallo et al., 1998* and *Fergnani et al., 2013*, we compared it to further occurrence datasets. The publication by *Ceccarelli et al., 2018* also contains comprehensive distribution data on triatomines. However, a direct comparison of both plotted datasets showed, that the data points obtained from *Ceccarelli et al., 2018* are completely covered by the data from the 'Atlas of the Chagas disease vectors in the Americas' (*Carcavallo et al., 1998*; *Fergnani et al., 2013*; *Supplementary file 6*). Furthermore, the occurrence data from the 'Atlas of Chagas disease vectors' (*Carcavallo et al., 1998*; *Fergnani et al., 2013*) match the time period of the climatic conditions used as predictor variables (1970–2000) and are probably less susceptible to sampling bias.

Additional global occurrences of *Triatoma rubrofasciata* from an intensive literature search were used solely for independent global model validation and were not included in the modelling approach (*Jurberg and Galvão, 2006*; *Eugenio and Minakawa, 2012*; *VAST, 2014*; *Dujardin et al., 2015b*; *Liu et al., 2017*; *Ceccarelli et al., 2018*; *Dong et al., 2018*; *Huang et al., 2018*; *GBIF. org, 2019a*). This type of global validation was only feasible for *T. rubrofasciata* as it is the only triatomine species with known occurrences both inside and outside the Americas. For the data collection, the search engine 'Google scholar' and 'Web of Knowledge' were searched for the keywords '*Triatoma rubrofasciata* occurrence', '*Triatoma rubrofasciata* distribution' and '*Triatoma rubrofasciata* records' considering only data points from outside the Americas. Records in English language, with included coordinates, and from all temporal periods were taken into account.

## Climate data

It is well described that temperature and relative humidity have a strong impact on the development and distribution of Triatominae favouring mild temperatures and median to high humidity (*Guarneri et al., 2003*; *Lazzari, 1991*; *Luz et al., 1998*; *Catalá et al., 2017*). Therefore, we proceeded on the assumption that the distribution of the Triatominae is mainly climatically controlled. Bioclimatic variables provided by WorldClim comprising data on temperature and precipitation patterns were used as environmental variables (*Fick and Hijmans, 2017*). Nineteen different variables are available referring to the climate conditions empirically recorded over a period of 30 years from 1970 to 2000. We chose a subset of six variables to train the models. Studies have indicated that a major limiting factor of triatomine distribution is the minimum temperature of the coldest month (*de la Vega et al., 2015*). However, this seems to be species-specific, since temperature seasonality has also often been identified as an important determinant (*Pereira et al., 2013*; *Ceccarelli et al.,*

*2015*). Hence, temperature seasonality (BIO4), maximum temperature of the warmest month (BIO5) and minimum temperature of the coldest month (BIO6) were chosen as explanatory variables for temperature. The precipitation and also the relative humidity play a decisive part in the distribution, but also the spatial delimitation of different triatomine species (*Gurgel-Gonçalves et al., 2011*; *Ibarra-Cerdeña et al., 2014*). Therefore, as explanatory variables for precipitation, we considered precipitation of the wettest month (BIO13), precipitation of the driest month (BIO14) and precipitation seasonality (BIO15). In order to avoid collinearity between the environmental variables, the Pearson correlation coefficient was computed (Pearson < 0.8) using the function *cor* of R's stats package (*R Development Core Team, 2013*).

## Species distribution modelling

The modelling of the habitat suitability was performed with an ensemble forecasting approach incorporating six different algorithms. Modelling was executed in the R environment (*R Development Core Team, 2019*) using the biomod2 package (*Thuiller et al., 2019*) (*Source code 1*). The algorithms were selected based on their modelling performance and advantages and included ANN – artificial neuronal networks, GAM – generalized additive models, GBM – generalized boosted models, GLM – generalized linear models, MARS – multivariate adaptive regression splines and MAXENT – maximum entropy approach (*Elith et al., 2006*; *Li and Wang, 2013*). The models were trained solely on the South American dataset with a spatial extent of 105°W to 35°W longitude and 30°N to 45°S latitude. The discriminatory capacity of the algorithms was evaluated using the receiver operating characteristic curve (ROC). A greater area under the curve (AUC 0–1) indicates a better predictive model performance. The results were then projected on a global scale. The models were run using the following single algorithm parameters: a stepwise feature selection with quadratic terms based on the Akaike Information Criterion (AIC) was used to generate the generalised linear models (GLM); generalised boosted models (GBM) were run with a maximum of 5 000 trees to ensure fitting, a minimum number of observations in trees' terminal nodes of 10, a learning rate of 0.01 and a interaction depth of 7; for generalised additive models (GAM) a binomial distribution and logit link function was applied and the initial degrees of smoothing was set to 4; the minimum interaction degree of the multivariate adaptive regression splines (MARS) was set to two with the number of terms to retain in the final model set to 17; artificial neuronal networks (ANN) were produced with fivefold cross-validation resulting in eight units in the hidden layer and a weight decay of 0.001; for the maximum entropy approach (MAXENT) we used linear, quadratic and product features and deactivated threshold and hinge features, while the number of iterations was increased to 10 000 to ensure convergence of the algorithm.

A background selection process was implemented and 10 000 pseudo-absences were chosen. For the presence/absence models ANN, GAM, GBM, GLM and MARS, we used the pseudo-absence selection parameter 'disk' which defines a minimal distance (80 km) to presence points for selecting pseudo-absence candidates. Since MAXENT is a presence/background modelling tool, the pseudo-absences were randomly chosen. Cross-validation of the models was carried out by splitting the dataset in a subset used for calibrating the models (70%) and a second subset to evaluate them (30%). All model predictions are scaled with a binomial GLM. This should lead to a reduction in projection scale amplitude and ensure comparable predictions. Consensus maps were built combining the modelling results of all algorithms with an AUC value >0.75. Their impact on the consensus maps was weighted by the mean of the AUC scores. Applying an ensemble forecasting approach yield a robust projection of the species' climate suitability.

We considered the relative contribution of the bioclimatic variables for all six algorithms and calculated their average importance over all models. Finally, the variables most contributing to the projection of the considered species were identified.

The modelled climatic suitability for eleven triatomine species was projected on a global extent under current climatic conditions. During model projection with biomod2, clamping masks were created. These masks signify areas where projections are uncertain because the values of the bioclimatic variables are outside the range used for calibrating the models. More precisely, this means that the models were trained with climatic variables whose values for temperature and precipitation correspond to the values occurring in South America. If the models are projected globally onto areas in which the temperature and precipitation patterns are outside the range of the trained values, extrapolation occurs and the projection can be regarded as uncertain. The clamping masks are

integrated into climatic suitability maps indicating areas where at least one variable exceeds their training range by hatching. Additionally, a detailed clamping mask is given in *Supplementary file 7*.

In order to convert the continuous climatic suitability maps into binary presence/absence maps, the equal sensitivity and specificity threshold was applied. Based on the binary modelling results, we compiled a diversity map combining all considered triatomine species. This merged map displays the number of species which find suitable climatic conditions in the respective grid cell on a global scale.

All maps were created with ESRI ArcGIS (*ESRI, 2018*).

## Model evaluation for *Triatoma rubrofasciata*

The evaluation of the global projection of climate suitability was performed with occurrence data of *T. rubrofasciata*, the only member of the *Triatoma* genus distributed in the Old and New World (*Dujardin et al., 2015b*). The global projection was compared to occurrence data of *T. rubrofasciata* outside of South America. Occurrence references with two levels of accuracy were taken into account; records with exact coordinates and records on country level. Occurrence points in areas with uncertain prediction have not been considered. Consistency of the global modelled climate suitability conditions and the actual occurrence of the species were compared and the resulting model performance was assessed.

## Acknowledgements

We thank Nicholas J Tobias (Senckenberg Gesellschaft für Naturforschung, Frankfurt am Main) for kindly revising the manuscript.

## Additional information

### Funding

No external funding was received for this work.

### Author contributions

Fanny E Eberhard, Conceptualization, Formal analysis, Visualization, Writing - original draft, Writing - review and editing; Sarah Cunze, Conceptualization, Supervision, Writing - review and editing; Judith Kochmann, Supervision, Writing - review and editing; Sven Klimpel, Conceptualization, Supervision, Funding acquisition, Writing - review and editing

### Author ORCIDs

Fanny E Eberhard (iD) https://orcid.org/0000-0002-4947-3867

### Decision letter and Author response

Decision letter https://doi.org/10.7554/eLife.52072.sa1
Author response https://doi.org/10.7554/eLife.52072.sa2

## Additional files

### Supplementary files

• Source code 1. Source code for species distribution modelling executed in the R environment (*R Development Core Team, 2019*) using the biomod2 package (*Thuiller et al., 2019*).

• Supplementary file 1. Modelled climatic suitability [%] for all occurrences of *T. rubrofasciata* outside of the Americas.

• Supplementary file 2. AUC values of all algorithms for all considered species.

• Supplementary file 3. Sensitivity and specificity metrics of the South American training dataset for all considered species and the independent global sensitivity analysis of *T. rubrofasciata*.

• Supplementary file 4. Modelled current climatic suitability and occurrence data from all considered species in South and Central America (*Carcavallo et al., 1998*; *Fergnani et al., 2013*).

• Supplementary file 5. Distribution of all considered species in South and Central America as extracted from the 'Atlas of Chagas disease vectors in America' (*Carcavallo et al., 1998*; *Fergnani et al., 2013*).

• Supplementary file 6. Juxtaposition of the occurrence data obtained from the 'Atlas of Chagas disease vectors' (*Carcavallo et al., 1998*; *Fergnani et al., 2013*) (black dots), *Ceccarelli et al., 2018* (blue triangles) and (*GBIF.org, 2019b*; *GBIF.org, 2019c*; *GBIF.org, 2019d*; *GBIF.org, 2019e*; *GBIF.org, 2019f*; *GBIF.org, 2019g*; *GBIF.org, 2019h*; *GBIF.org, 2019i*; *GBIF.org, 2019j*; *GBIF.org, 2019k*; *GBIF.org, 2019l*) (red triangles) of all considered species.

• Supplementary file 7. Clamping mask indicating areas in which one or more environmental variables are outside the values of their training range. The climatic suitability projections in these areas can be regarded as uncertain.

• Supplementary file 8. Sensitivity and specificity metrics of all algorithms for all considered species.

• Transparent reporting form

## Data availability

All data generated or analysed during this study are included in the manuscript and supporting files.

The following previously published datasets were used:

| Author(s) | Year | Dataset title | Dataset URL | Database and Identifier |
|---|---|---|---|---|
| GBIF | 2019 | GBIF Occurrence Download | http://doi.org/10.15468/dl.yneo2v | Global Biodiversity Information Facility, 10.15468/dl.yneo2v |
| Fergnani PN, Ruggiero A, Ceccarelli S, Menu F, Rabinovich J | 2013 | Large-scale patterns in morphological diversity, and species assembly in Neotropical Triatominae (Heteroptera: Reduviidae) | https://doi.org/10.6084/m9.figshare.653959.v6 | figshare, 10.6084/m9.figshare.653959.v6 |

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
