## [Decision Letter]

**Acceptance summary:**

This paper provides an interesting analysis of the climate and environmental factors driving the vector occurrence for Chagas disease, a tropical disease that is transmitted by insects of the Triatominae subfamily and affects several million people worldwide. The topic and the results of this research are relevant, novel and with important public health implications for a global vector surveillance effort.

**Decision letter after peer review:**

Thank you for submitting your article "Modelling the climatic suitability of Chagas disease vectors on a global scale" for consideration by *eLife*. Your article has been reviewed by Neil Ferguson as the Senior Editor, a Reviewing Editor, and two reviewers. The following individuals involved in review of your submission have agreed to reveal their identity: Zulma Cucunubá (Reviewer #2).

The reviewers have discussed the reviews with one another and the Reviewing Editor has drafted this decision to help you prepare a revised submission.

Summary:

This paper provides an interesting analysis of the climate and environmental drivers of Chagas vector occurrence, and comments on the implications for surveillance.

Both reviewers found the paper of interest but identified major limitations which need to be addressed. Given the nature of some comments, acceptance of a revised manuscript is not guaranteed.

Essential revisions:

Refer to the full reviews for details but the following are the most critical issues:

- the source data – both reviewers comment that it is not very current. More recent occurrence data should be included if at all possible. Why was the Cecarelli, 2018 dataset not used? More detail on the data is also needed (see reviewer 2).

- pseudo absence points – comments of reviewer 1 need to be fully addressed, including the lack of absence points in the validation dataset.

- validation set – data from only one species was used (see reviewer 1) – again, the rationale for this needs to be given, and I would prefer to see spatially stratified model selection and cross-validation used (e.g. spatial block bootstrap).

- model choice and settings used – these need to be justified and sensitivity analyses undertaken (see reviewer 1). In general, more detail of the modelling (including the ensemble approach) is needed – see both reviews.

Reviewer #1:

Overall, an interesting topic with some relevant approaches applied. Currently some serious lack of detail and rigour in the modelling approach that prevents this from being a valuable addition to the literature and may not be feasible to address in reasonable timescales. This makes the results and their significance difficult to interpret.

Essential revisions:

The occurrence data from these models come from a single source published in 1998 (Carcavallo et al., 1998). Surely there must be more up-to-date data on occurrence of these species? Particularly with the advent of services like GBIF (which the authors cite for one species). The robustness of these maps could be substantially improved if more modern data were included.

The "validation set" is comprised of data from a literature review for *Triatoma rubrofasciata* and appears to cover a more modern time period (citations dated 2006-2009). Why was this only done for one species? Doing a prospective evaluation of the ENM is certainly one approach, but the limitations of this validation approach should be explored, e.g. confirmation bias (are people just doing surveys in areas where the atlas indicates presence?), important changes in the distribution over time, etc, etc.

Pseudo-absence and lack of absence data in validation set. The choice of random pseudo absence generation when combined with non-systematically sampled occurrence data is problematic for both accuracy metrics and overprediction and has been discussed at length (e.g. Chefaoui and Lobo, 2008) – effectively it means you map surveillance effort not occurrence of the species. I don't think random pseudo absence data is a suitable choice for this approach given the variable surveillance effort. Also including Arctic and Antarctic areas and generally areas that are a long way away from presence points is a good way to artificially boost your AUC – is anyone really hypothesising that these species can spread to these regions? The lack of absence data in the "validation set" is also problematic and leads the models to prioritise sensitivity over specificity. Arguably this should be the other way around as the primary use for the maps is to target surveillance to areas where importation may be a problem. The authors should consider a more nuanced approach to absence data. Could occurrence points for other species be indicative of surveillance effort?

Subsection “Species distribution modelling”, "All algorithms were run with default settings" – these are a complex set of methods with a large number of tuneable hyperparameters. I'm not sure it is a fair comparison to just leave them with default settings, nor is it a good way to optimise fit. I'd like to see a clear rationale for why these classes of methods were chosen, relevant choices for hyperparameters and ideally some experiments to validate these choices.

Reviewer #2:

The manuscript reports a niche modelling predicting the global climatic suitability of eleven triatomine species (competent *Trypanosoma cruzi* vector).

The topic and the results of this research are very relevant, novel and with important public health implications for a global vector surveillance effort, especially regarding *Triatoma rubrofasciata*. The paper also provides a comprehensive discussion. However, the manuscript lacks some methodological details to help the reader understand how the analysis was conducted and which are the limitations and implications of both the methodological approach and therefore the results.

Occurrence records:

The section describing the data should be extended. I acknowledge the authors have used a dataset of 4155 unique points, most of them already collated by other authors (Fergnani et al., 2013) who in turn extracted the data from another publication (Carcavallo et al., 1998). But some basic information is needed in order to assess what the dataset encompasses.

Was there any quality control used for data extraction?

Are there any concerns about biases in data collection?

Did the data undertake any time standardisation?

What are the potentials concerns of not having occurrence data beyond 1999 for the American triatomine species?

It would be important to have a figure showing the distribution of the data points per species, even if it is just on the Supplementary materials.

Very important, some of the authors from the main source of information (Fergnani et al., 2013), published a data paper (Ceccarelli et al., 2018) with 21815 georeferenced triatomine records updated until 2017. What are the implications of using (or not using) a more updated dataset like this one?

Results section:

I acknowledge this is a prediction effort at a global scale, but I found hard to understand how the model predicts about 70% of climatic suitability for some species across very large areas that include the highlands (above 2500 MAMSL) in South America (i.e. *R. prolixus* or *P. geniculatus* in Bogota). This even considering the model does not include a climate change scenario. Also, when compared to previous publications on climatic suitability in the Americas I found concerning differences for some species such as *R. prolixus* in Colombia (Parra-Henao et al., 2016) or *P. megistus* in Brazil (Gurgel-Gonçalves et al., 2012).

From the maps it seems this work tends to predict a much larger distribution of some of these species in the Americas than previous works did. Not having high resolution maps and the absence of country boundaries makes it harder to tell about potential problems in the predictions at a smaller scale.

It is interesting to see that although for some species the number of records is very scarce (i.e. *Rhodnius ecuadoriensis* n = 31) the AUC values are still very high. There is no mention about the limitations regarding the data on the Discussion section.

On Figure 1A 'consensus model' is mentioned. But the basic details about this model have not been mentioned on the Materials and methods section.

Discussion section:

The authors mention "we were able to divide the considered species roughly into three groups dependent on their climatic habitat preferences". I did not find clearly which are those three groups and which were the methods to identify them.

To put this work into context, it would be important to include a discussion point about the highly effective vector control and other factors (i.e. housing conditions) that would potentially determine environmental suitability, beyond the climatic suitability.

[Editors’ note: further revisions were suggested prior to acceptance, as described below.]

Thank you for submitting your article "Modelling the climatic suitability of Chagas disease vectors on a global scale" for consideration by *eLife*. Your article has been reviewed by Anna Akhmanova as the Senior Editor, a Reviewing Editor, and two reviewers. The reviewers have opted to remain anonymous.

The reviewers have discussed the reviews with one another and the Reviewing Editor has drafted this decision to help you prepare a revised submission. In recognition of the fact that revisions may take longer than the two months we typically allow, until the research enterprise restarts in full, we will give authors as much time as they need to submit revised manuscripts.

The reviewers agreed that the paper has been substantially improved, but also identified some remaining points that need to be addressed. No collection of new data will be needed to address reviewer comments.

Essential revisions:

1) Please add additional detail about the data used, how they were extracted, curated and filtered prior to analysis to the Materials and methods section of the manuscript.

2) Running the models with default parameter values only – both reviewers felt this point was insufficiently addressed. Please conduct further dataset-specific analyses to support your choice of model parameters.

3) Please review the manuscript figures to make it clearer how well the model prediction matches the data and be more explicit how uncertainty was calculated and represented and include this in the main text when discussing findings.

Please see below the individual comments from each reviewer for a more detailed explanation of issues related to each of the above points. All reviewer points will need to be addressed point-by-point in your revised submission.

Reviewer #1:

I'd like to thank the authors for their detailed responses and additions to this work in regards to the majority of my points raised. I think all but one of these have now been adequately addressed. On point 6 [running models with default parameters only] – I don't think this particular comment has been addressed. Suggesting that such parameters have been "optimised by the biomod2 development team" is not realistic given the breadth of problems that these algorithms are applied to. To take one example, in the documentation for GAMs in the "mgcv" package (that biomod2 calls) there is extensive advice on basis dimension choice for smooths and the explicit statement "The choice of the basis dimension (k in the s, te, ti and t2 terms) is something that should be considered carefully" and a range of model diagnostic statistics and plots are suggested to tune such parameters. This is one example of many and, as a reader, I do not have great confidence in the work if some of these model flexibility parameters are not at least explored. What makes the issue worse is that a reader currently has no way of diagnosing what impact this oversight might have as there are no model coefficients or effects plots presented in the manuscript. I appreciate that this is a common oversight in many ML modelling applications, but even a basic sensitivity analysis would be a big improvement over using the default values.

Reviewer #2:

I acknowledge the authors have made substantial improvements to the original version of the manuscript following the reviewers' recommendations. The modifications imply a remarkable change on the original predicted distributions. However, some considerations in terms of the methodology and the presentation of the results remain.

About the data:

My main worry is that the methods section remains limited in the details and particularly in terms of the data that has been used, which makes very difficult to understand all the work that has been done. I suggest the authors consider adding a sub-section on the Materials and methods section dedicated exclusively to explain where the data come from.

For example, the authors mention as data source the "Atlas of Chagas disease vectors in the Americas (Carcavallo et al., 1998) which were digitised at a 0.1^o^ x 0.1^o^ resolution by Fergnani et al. (2013)". What does exactly "digitised" mean? Is it Fergnani already a modelling work on the Atlas data? What is the difference between Carcavallo and Fergnani data? This becomes even more important as Carcavallo is a book with restricted access so that it is difficult to trace the original source.

This is further confusing later when the authors cite Supplementary file 4 as the occurrence data, citing Carcavallo and not Fergnani.

In subsection “Occurrence data” they mention that "In total, 4155 unique occurrence points were collected ranging from 31 for *Rhodnius ecuadoriensis* to 1180 for *Panstrongylus geniculatus* (Table 1)." Were these points collected by the authors? This is somehow contradictory to the use of already collected data from Carcavallo/Fergnani.

Further on the same topic, the authors mention on their reply to the reviewers that they have "We carefully compared both datasets and plotted them in ArcGIS. It turned out that the Ceccarelli as well as the GBIF occurrence records are completely covered by the Atlas data". This should be explicitly mentioned in the Materials and methods section and the comparison map added as supplementary information.

Also, the authors mention (subsection “Occurrence data”) that "Additional global occurrences of *Triatoma rubrofasciata* from an intensive literature search were used". However, in the Materials and methods section there is not mention to the details of the review process followed to obtain such data (which databases, which quality control, which languages, which temporal filter they have used, etc). If the data for *Triatoma rubrofasciata* is used as data points, how different is the methodology for this species compared to the other species?

About the statistical methods:

In subsection “Species distribution modelling”, the authors mention "All algorithms were run with default settings except for MAXENT, GLM and GBM." In response to a reviewer's comment about what those default setting imply, the authors mention that "We have carefully examined the different parameters and changed the information criteria for the stepwise selection procedure in GLM to 'Akaike Information Criteria (AIC)' and the number of terminal nodes in GBM to 6 as it is recommended by Friedman (2002)". I believe the reasoning behind the "default settings" has not been clarified yet.

About the Results section and Discussion section:

In Supplementary file 4 there is not needed to show the background colours but simply the distribution of the data. The background does not really help to see the data.

Could you please explain why in the Global validation it was possible to estimate sensitivity but not specificity for *T. rubrofasciata*?

In the Results section it is mentioned several times some agreements and disagreements between the model and the data for various species. For example, in the Discussion section "the models appear to slightly overestimate the potential distribution as it could be noted in the modelling of *T. dimidiate*". However, it is actually hard for the reader to note exactly where these potential overestimates are occurring. It will be great if you can have a figure (even if it is a set of figures in Supplementary file) where you show both the model predictions with the occurrence data on top so the reader can judge and understand where the model is fitting well and not that well, as you have done for *T. rubrofasciata* on Figure 2.

In Figure 1 (and also Figure 2) it is mentioned that "Hatched areas indicate regions where the projection is uncertain". There are two problems with this uncertainty:

- The size of the panels makes the figures so small that it is impossible to actually see the hatched areas.

- What does it mean "uncertain"? It should be clearly explained in the Materials and methods section how such uncertainty was estimated. Is there a metric for such uncertainty?

These problems with showing uncertainty in both Figure 1 and Figure 2 could be solved by having other similar figures exclusively for uncertainty.

To avoid confusion, I encourage authors to use a more cautious language when referring to climate suitability rather than actual presence of a particular species. For example in subsection “Potential distribution under current climate conditions” they mention "*T. brasiliensis* prefers dry and wet savannah climate as found in eastern Brazil and southern West Africa, northern and southern Central Africa and East Africa". But, in reality *T. brasiliensis* hasn't ever been found in Africa.

---

## [Author Response]

Essential revisions:Refer to the full reviews for details but the following are the most critical issues:- the source data – both reviewers comment that it is not very current. More recent occurrence data should be included if at all possible. Why was the Cecarelli, 2018 dataset not used? More detail on the data is also needed (see reviewer 2).- pseudo absence points – comments of reviewer 1 need to be fully addressed, including the lack of absence points in the validation dataset.-validation set – data from only one species was used (see reviewer 1) – again, the rationale for this needs to be given, and I would prefer to see spatially stratified model selection and cross-validation used (e.g. spatial block bootstrap).- model choice and settings used – these need to be justified and sensitivity analyses undertaken (see reviewer 1). In general, more detail of the modelling (including the ensemble approach) is needed – see both reviews.

Before answering point by point, we would like to briefly summarise our answers to the main issues:

A) Occurrence data used for modelling

There are generally two potential datasets that can be used as occurrence records and possess different advantages and drawbacks; Carcavallo et al. (1998), which were digitised at a 0.1° x 0.1° resolution by Fergnani et al. (2013) and used in our first version, and point data from e.g. Ceccarelli et al. (2018) and GBIF (Global Biodiversity Information Facility). We carefully compared both datasets and plotted them in ArcGIS. It turned out that the Ceccarelli as well as the GBIF occurrence records are completely covered by the Atlas data. We therefore do not assume that the Atlas data underestimate the distribution of the vectors.

Furthermore, we decided to use the Atlas data for the following reasons:

- the Atlas data match the time period of the climatic conditions used as predictor variables (WorldClim version 2, 1970-2000).

- point data such as GBIF data reflect the sampling and surveillance effort in a considered area, while the Atlas data are probably less effected by sampling bias.

- it also follows that the selected pseudo-absences are more reliable.

B) Background selection

We agree that the random background selection is not appropriate for all applied algorithms. We thus reran the analysis and now differentiate between presence-background models (MAXENT) and presence-absence models (ANN, GAM, GBM, GLM and MARS). During modelling with MAXENT, we still selected the background randomly as it is required. During modelling with the presence-absence algorithms, we selected the background (to be interpreted as pseudo-absences) using two different stratified sampling methods and compared the results. In the end, we decided to choose a background selection with geographical filtration (‘disk’) and not environmental filtration (‘sre’), because the latter tends to overestimate the distribution of the species (Figure 1).

C) Model validation

*Triatoma rubrofasciata* is the only triatomine species with recorded occurrences outside the Americas and in Latin America. Therefore, in our first version, we used this species to independently evaluate the global projection of the models comprising a sensitivity analysis. In the revised version, we also applied a cross-validation by splitting the dataset in a training set (70%) and a test set (30%).

D) More details on Materials and methods

We carefully reworked the Material and methods section, added more details to this part and expanded on some points in the Discussion section.

Reviewer #1:Overall, an interesting topic with some relevant approaches applied. Currently some serious lack of detail and rigour in the modelling approach that prevents this from being a valuable addition to the literature and may not be feasible to address in reasonable timescales. This makes the results and their significance difficult to interpret.Essential revisions:The occurrence data from these models come from a single source published in 1998 (Carcavallo et al., 1998). Surely there must be more up-to-date data on occurrence of these species? Particularly with the advent of services like GBIF (which the authors cite for one species). The robustness of these maps could be substantially improved if more modern data were included.

We agree with reviewer 1 that data acquisition is quite often a challenge and at the same time the most crucial aspect for modelling. We are aware of the publication of recent occurrence data, in particular of the paper Ceccarelli et al. (2018) and the data from GBIF. We compared the recorded distribution of both datasets in ArcGIS and found that the distribution of the occurrence records from Ceccarelli as well as from GBIF are completely covered by the data of the Atlas (Carcavallo et al., 1998). This suggests that in recent years there has been no significant range expansion despite changes in climate. It could further indicate that the different species have stable distributions in South America.

Furthermore, the environmental variables from WorldClim comprise a time period from 1970 to 2000. Unfortunately, newer comprehensive climatic data are not available. Therefore, the Atlas data fit the explanatory climate variables much better than the newer GBIF and Ceccarelli distribution data. In addition, the comparison of the data, as already described, has not shown any noteworthy distribution shifts in recent years.

We would also like to point out that the Atlas data do not represent single point data, but have been digitized from maps in the Atlas (Carcavallo et al., 1998). This greatly mitigates a potential sampling or surveillance bias, which is evidently present in the Ceccarelli data as well as the GBIF data. This becomes clear, for example, in *P.geniculatus*, which is widespread in humid tropical forests to tropical dry forests and savannah in Colombia, Venezuela, Trinidad, Guyana, Surinam, French Guyana, Brazil, Ecuador, Peru, Bolivia, Paraguay, Uruguay, Argentina, Mexico, Guatemala, Nicaragua, Costa Rica and Panama (Patterson et al., 2009). The Ceccarelli and GBIF data cover many occurrence points in Colombia, French Guiana and in the state of Espírito Santo in Brazil, where a ten-year entomological surveillance program was applied (Leite et al., 2007). In the Amazon Basin, however, the availability of occurrence points is very scarce and all registered points are located along river branches. Furthermore, the GBIF as well as the Ceccarelli data underestimate the species distribution achieved, especially for *Triatoma rubrofasciata*. For example, the vector has been detected in the Caribbean and in Central and North America which is not reproduced in the GBIF nor the Ceccarelli data sets, but in the Atlas data sets.

Due to a lower sampling bias, the Atlas data are also more suitable for presence-absence (PA) models such as GLM, GBM, GAM, ANN and MARS since the calculated pseudo-absence points are more reliable.

The "validation set" is comprised of data from a literature review for Triatoma rubrofasciata and appears to cover a more modern time period (citations dated 2006-2009). Why was this only done for one species? Doing a prospective evaluation of the ENM is certainly one approach, but the limitations of this validation approach should be explored, e.g. confirmation bias (are people just doing surveys in areas where the atlas indicates presence?), important changes in the distribution over time, etc, etc.

*Triatomarubrofasciata* is the only triatomine species found both in the Americas and in the Old World and has already spread to various countries. Therefore, it is the only species suitable for independent model validation outside the Americas. Due to very limited data, occurrence points of *T.rubrofasciata* from 1958 to 2018 were collected and used for validation. It is true that this results in certain limitations that need to be addressed. We have adjusted our Discussion section accordingly.

Also, a manual sensitivity analysis of the *T. rubrofasciata* modelling results was performed. For this purpose, the modelled climatic habitat suitability of *T.rubrofasciata* was read from global occurrence points and a cut-off of 50.95% was applied. This cut-off was obtained from the calculation of the ROC. The number of ‘true positives’ in the total amount of presence data was determined. This resulted in a sensitivity of 66.6%.

Additionally, we applied a cross-validation of the South American training data set.

Pseudo-absence and lack of absence data in validation set. The choice of random pseudo absence generation when combined with non-systematically sampled occurrence data is problematic for both accuracy metrics and overprediction and has been discussed at length (e.g. Chefaoui and Lobo, 2008) – effectively it means you map surveillance effort not occurrence of the species. I don't think random pseudo absence data is a suitable choice for this approach given the variable surveillance effort.

We thank reviewer 1 for this valuable advice and fully implemented it. A stratified background selection is strongly required when using presence-absence models but a random selection is appropriate when using presence-background models. We account for this in the revised analysis and used a stratified background sampling (with two different strategies: ‘disk’ and ‘sre’, for details see below) for the presence-absence models (GLM, GBM, GAM, ANN and MARS) but retained the random background selection for MAXENT.

We used both the pseudo-absence selection option ‘sre’ and the option ‘disk’ and compared them to one another. For the option ‘sre’ a surface range envelope model is first carried out (using a specified quantile 2.5%) on the species of interest, and then the pseudo-absence data are extracted outside of this envelope broadly comprising the environmental conditions for the species. This avoids absence selection in the same niche as the niche of the considered species. For the option ‘disk’, a minimal distance to presence points (80km, diagonal between two occurrence points) for selecting pseudo-absences candidates is defined. This avoids absence selection close to observed presences, and thus with similar climatic conditions. In the end, we opted for the absence selection with parameter ‘disk’ (Materials and methods section), because models with an absence selection with a surface range envelope tend to project very large areas as climatic suitable for the considered species – a circumstance that we could also observe in part in our models (Figure 1).

MAXENT is a presence-background modelling tool applying the principle of maximum entropy in such a way that the estimated species distribution deviates from a uniform distribution as little as it is necessary to explain the observations. Therefore, ‘random’ background selection is still required (nicely explained in Elith et al., 2011). Finally, we built the consensus maps from all six models. It is probably advisable that ensemble models in which presence-absence and presence-background models are used simultaneously, a nuanced pseudo-absence selection approach should always be used. Unfortunately, assembling all models requires a lot of manual work.

Also including Arctic and Antarctic areas and generally areas that are a long way away from presence points is a good way to artificially boost your AUC – is anyone really hypothesising that these species can spread to these regions?

We agree with the reviewer that an AUC validation applied to a broad range of environmental conditions can artificially improve the AUC outcome. However, the AUC values we provide are based only on the South American data (the training set) and do not include these regions. Only the projections of the potential distributions were carried out at a global scale. Nevertheless, we have now removed the Antarctic region for the depiction of the projection results (Figure 1) and tried to describe the process of modelling more clearly (Materials and methods section).

The lack of absence data in the "validation set" is also problematic and leads the models to prioritise sensitivity over specificity. Arguably this should be the other way around as the primary use for the maps is to target surveillance to areas where importation may be a problem. The authors should consider a more nuanced approach to absence data. Could occurrence points for other species be indicative of surveillance effort?

We followed the reviewer’s suggestion and adapted the validation of the models. It was carried out by splitting the original Latin American dataset into a subset used for calibrating the models (70% of the dataset) and a second subset to evaluate them (30%, cross-validation) (Materials and methods section).

For the manual global validation with *T.rubrofasciata*, we cannot provide meaningful absence data due to potential dispersal of the species.

The distribution of all South American GBIF data could be used to map the surveillance effort and to incorporate this into the modelling as a correction factor. This is a very interesting method, but not feasible here, since the Atlas data are not point data.

Subsection “Species distribution modelling”, "All algorithms were run with default settings" – these are a complex set of methods with a large number of tuneable hyperparameters. I'm not sure it is a fair comparison to just leave them with default settings, nor is it a good way to optimise fit. I'd like to see a clear rationale for why these classes of methods were chosen, relevant choices for hyperparameters and ideally some experiments to validate these choices.

We have carefully examined the different parameters and changed the information criteria for the stepwise selection procedure in GLM to ‘Akaike Information Criteria (AIC)’ and the number of terminal nodes in GBM to 6 as it is recommended by Friedman, (2002) (Materials and methods section). The other settings in biomod2 have already been optimized by the biomod2 development team and are not consistent with the default settings of the individual algorithms. However, if you have specific suggestions, we are happy to implement them.

Reviewer #2:The manuscript reports a niche modelling predicting the global climatic suitability of eleven triatomine species (competent Trypanosoma cruzi vector).The topic and the results of this research are very relevant, novel and with important public health implications for a global vector surveillance effort, especially regarding Triatoma rubrofasciata. The paper also provides a comprehensive discussion. However, the manuscript lacks some methodological details to help the reader understand how the analysis was conducted and which are the limitations and implications of both the methodological approach and therefore the results.Occurrence records:The section describing the data should be extended. I acknowledge the authors have used a dataset of 4155 unique points, most of them already collated by other authors (Fergnani et al., 2013) who in turn extracted the data from another publication (Carcavallo et al., 1998). But some basic information is needed in order to assess what the dataset encompasses.Was there any quality control used for data extraction?

We agree with the reviewer that a detailed quality control would be appropriate if the distribution data were point data from different sources, such as GBIF. Accordingly, it would be reasonable to sort out points that are close to one another, uncertain data, very old or new data that do not fit the explanatory climate variables. However, this is not useful for the Atlas data, as these are digitised distribution maps. For further details regarding the distribution data see point A ‘Occurrence data used for modelling’.

Are there any concerns about biases in data collection?

A sampling bias should not be very pronounced due to the digitised distribution maps and the lack of point occurrences from different sources.

Did the data undertake any time standardisation?

As part of the revision, we dealt a lot with the comparison of Atlas data and point data from GBIF and the literature and have considered this point in more depth. The Atlas data from 1998 fit well with the period in which the climate data (1970-2000) were recorded.

What are the potentials concerns of not having occurrence data beyond 1999 for the American triatomine species?

Since we train the models with climate data from 1970 to 2000, it is advisable not to use any occurrence data after 2000. It is possible and very likely that the climate has changed since then and that the modelled habitat suitability would look slightly different under present conditions. Particularly in the course of climate change, the models might underestimate the current distribution of thermophilic species. We included this point in the discussion (Discussion section). All in all, it would be good to have more recent and up-to-date climatic data, but these have not (yet) been published.

It would be important to have a figure showing the distribution of the data points per species, even if it is just on the Supplementary materials.

Done as suggested. We included distribution maps in the supplementary materials.

Very important, some of the authors from the main source of information (Fergnani et al., 2013), published a data paper (Ceccarelli et al., 2018) with 21815 georeferenced triatomine records updated until 2017. What are the implications of using (or not using) a more updated dataset like this one?

The issue was also mentioned by reviewer 1 and we combined our reply. For detailed explanation, please refer to point A “Occurrence data used for modelling”.

Results section:I acknowledge this is a prediction effort at a global scale, but I found hard to understand how the model predicts about 70% of climatic suitability for some species across very large areas that include the highlands (above 2500 MAMSL) in South America (i.e. R. prolixus or P. geniculatus in Bogota). This even considering the model does not include a climate change scenario. Also, when compared to previous publications on climatic suitability in the Americas I found concerning differences for some species such as R prolixus in Colombia (Parra-Henao et al., 2016) or P. megistus in Brazil (Gurgel-Gonçalves et al., 2012).From the maps it seems this work tends to predict a much larger distribution of some of these species in the Americas than previous works did. Not having high resolution maps and the absence of country boundaries makes it harder to tell about potential problems in the predictions at a smaller scale.

This could be due to the coarse resolution of the used occurrence data (rasterised polygon data). By applying the changed pseudo-absence selection, the modelled potential distribution has decreased. The algorithms work now more discriminatory, especially in higher altitudes. This can be seen, for example, in the different models for *Rhodnius prolixus* and *Triatoma dimidiata* (Figure 1). Furthermore, this work strives to indicate areas at risk on a global scale where triatomine species find climatically suitable environments. Therefore, we tried to use the algorithms not too restrictively.

High resolution maps (500dpi) are now available in the supplementary materials and administrative borders are included in the maps.

It is interesting to see that although for some species the number of records is very scarce (i.e. Rhodnius ecuadoriensis n = 31) the AUC values are still very high. There is no mention about the limitations regarding the data on the Discussion section.

This is not surprising, on the contrary, because of the few occurrence points, the AUC value is high for this species. The AUC value depends on the prevalence and, therefore, cannot be compared between species, but can only be used to compare the discriminatory quality of different models for a single species.

On Figure 1A 'consensus model' is mentioned. But the basic details about this model have not been mentioned on the Materials and methods section.

‘Consensus model’ refers to the ‘consensus maps’ of the ensemble forecasting described in the Materials and methods section. We harmonised our wording so that this becomes clearer.

Discussion section:The authors mention "we were able to divide the considered species roughly into three groups dependent on their climatic habitat preferences". I did not find clearly which are those three groups and which were the methods to identify them.

This was a rough classification based on modelled habitat preferences as it is described in the Results section. We acknowledge that this apportionment into groups might be misleading and therefore opted for its removal (Results section; Discussion section).

To put this work into context, it would be important to include a discussion point about the highly effective vector control and other factors (i.e. housing conditions) that would potentially determine environmental suitability, beyond the climatic suitability.

A good point which we missed to mention in the first version. We included information about vector control measurements (Discussion section).

[Editors’ note: further revisions were suggested prior to acceptance, as described below.]

Essential revisions:1) Please add additional detail about the data used, how they were extracted, curated and filtered prior to analysis to the Materials and methods section of the manuscript.

We have completely revised the subsection “Data collection” and tried to explain the processing of the occurrence data more precisely. We also go into more detail about the compilation of the *Triatoma rubrofasciata* dataset.

2) Running the models with default parameter values only – both reviewers felt this point was insufficiently addressed. Please conduct further dataset-specific analyses to support your choice of model parameters.

A tuning step of the individual algorithm parameters was conducted using the BIOMOD_tuning function of the R package biomod2. The optimised parameters were then used to perform a new modelling analysis.

3) Please review the manuscript figures to make it clearer how well the model prediction matches the data and be more explicit how uncertainty was calculated and represented and include this in the main text when discussing findings.

We have added several supplementary figures, which compare the modelling results with the occurrence of the species as well as the different datasets. An additional figure of the modelling uncertainty has also been attached and addressed in the main text.

Reviewer #1:I'd like to thank the authors for their detailed responses and additions to this work in regards to the majority of my points raised. I think all but one of these have now been adequately addressed. On point 6 [running models with default parameters only] – I don't think this particular comment has been addressed. Suggesting that such parameters have been "optimised by the biomod2 development team" is not realistic given the breadth of problems that these algorithms are applied to. To take one example, in the documentation for GAMs in the "mgcv" package (that biomod2 calls) there is extensive advice on basis dimension choice for smooths and the explicit statement "The choice of the basis dimension (k in the s, te, ti and t2 terms) is something that should be considered carefully" and a range of model diagnostic statistics and plots are suggested to tune such parameters. This is one example of many and, as a reader, I do not have great confidence in the work if some of these model flexibility parameters are not at least explored. What makes the issue worse is that a reader currently has no way of diagnosing what impact this oversight might have as there are no model coefficients or effects plots presented in the manuscript. I appreciate that this is a common oversight in many ML modelling applications, but even a basic sensitivity analysis would be a big improvement over using the default values.

Based on the valuable suggestions of both reviewers, we decided to rerun the modelling and completely revise the selection of the algorithm parameters. For this purpose, we used the internal function BIOMOD_tuning from the biomod2 package. This function was designed to tune the biomod single models parameters based on the optimisation of the AUC values and returns a ModelingOptions object which can be used for modelling. The following optimised parameters where then used to perform a new modelling approach: (1) a stepwise feature selection with quadratic terms based on the Akaike Information Criterion (AIC) was used to generate the generalised linear models (GLM); (2) generalised boosted models (GBM) were run with a maximum of 5 000 trees to ensure fitting, a minimum number of observations in trees’ terminal nodes of 10 as we have large training datasets, a learning rate of 0.01 and a interaction depth (maximum number of nodes per tree) of 7; (3) for generalised additive models (GAM) a binomial distribution and logit link function was applied, the initial degrees of smoothing was set to 4; (4) the minimum interaction degree of the multivariate adaptive regression splines (MARS) was set to 2 with the number of terms to retain in the final model set to 17; (5) artificial neuronal networks (ANN) were produced with fivefold cross-validation resulting in 8 units in the hidden layer and a weight decay of 0.001; for the maximum entropy approach (MAXENT) we used linear, quadratic and product features and deactivated threshold and hinge features, the number of iterations was increased to 10 000 (Materials and methods section).

The use of the adjusted parameters did not result in major changes in the modelling output (see Figure 1). It can therefore be assumed that an ensemble forecasting approach with several algorithms produces results that are rather robust against changes in the single algorithm parameters. However, we opted to include the new modelling results in the manuscript since the AUC values were improved by the parameter optimisation indicating a better predictive model performance.

Reviewer #2:I acknowledge the authors have made substantial improvements to the original version of the manuscript following the reviewers' recommendations. The modifications imply a remarkable change on the original predicted distributions. However, some considerations in terms of the methodology and the presentation of the results remain:About the data:My main worry is that the Materials and methods section remains limited in the details and particularly in terms of the data that has been used, which makes very difficult to understand all the work that has been done. I suggest the authors consider adding a sub-section on the Materials and methods section dedicated exclusively to explain where the data come from.For example, the authors mention as data source the " the 'Atlas of Chagas disease vectors in the Americas' (Carcavallo et al., 1998) which were digitised at a 0.1^o^ x 0.1^o^ resolution by Fergnani et al. (2013)". What does exactly "digitised" mean? Is it Fergnani already a modelling work on the Atlas data? What is the difference between Carcavallo and Fergnani data? This becomes even more important as Carcavallo is a book with restricted access so that it is difficult to trace the original source.This is further confusing later when the authors cite Supplementary file 4 as the occurrence data citing Carcavallo and not Fergnani.In subsection “Occurrence data” they mention that "In total, 4155 unique occurrence points were collected ranging from 31 for Rhodnius ecuadoriensis to 1180 for Panstrongylus geniculatus (Table 1)." Were these points collected by who, by the authors? This is somehow contradictory to the use of already collected data from Carcavallo/Fergnani.

We have completely revised the subsection “Occurrence data” and explain the collection and processing of the data in more detail. In particular, we discuss the conversion of the distribution maps from the Atlas into point data by Fergnani et al. (2013) (Materials and methods section):

“Occurrence data of the triatomine species were obtained from data provided by Fergnani et al. (2013). […] With the help of the map projection, occurrence points with coordinates were created (Fergnani et al., 2013).”

Furthermore, since both Carcavallo et al. (1998) and Fergnani et al. (2013) are equally relevant for providing the occurrence data, we have decided to always cite both sources.

Further on the same topic, the authors mention on their reply to the reviewers that they have "We carefully compared both datasets and plotted them in ArcGIS. It turned out that the Ceccarelli as well as the GBIF occurrence records are completely covered by the Atlas data". This should be explicitly mentioned in the Materials and methods section and the comparison map added as supplementary information.

Done as suggested. We explicitly mentioned the comparison of the Atlas data and the Ceccarelli and GBIF data in the Materials and methods section, explained the choice of the Atlas dataset (Materials and methods section) and added a set of figures (Supplementary file 6) showing occurrence data from the ‘Atlas of Chagas disease vectors in the Americas’ (Carcavallo et al., 1998), occurrence data from Ceccarelli et al. (2018) and occurrence data from GBIF.org (2019).

Also, the authors mention (subsection “Occurrence data”) that "Additional global occurrences of Triatoma rubrofasciata from an intensive literature search were used". However, in the Materials and methods section there is not mention to the details of the review process followed to obtain such data (which databases, which quality control, which languages, which temporal filter they have used, etc). If the data for Triatoma rubrofasciata is used as data points, how different is the methodology for this species compared to the other species?

The occurrence data for the modelling approach of all species were solely obtained from the ‘Atlas of Chagas disease’ (Carcavallo et al., 1998) and more specifically the publication by Fergnani et al. (2013) which covers only the Americas. Individual records from other sources were not included.

The independent global model validation was only feasible with *T.rubrofasciata* as it is the only triatomine species with known occurrence data in the Americas and outside the Americas. Therefore, additional sources had to be used as the Atlas provides no records outside of the Americas.

However, we now define both datasets in more detail and describe the collection of the occurrence records of *T.rubrofasciata* (Materials and methods section).

About the statistical methods:In subsection “Species distribution modelling”, the authors mention "All algorithms were run with default settings except for MAXENT, GLM and GBM.". In response to a reviewer's comment about what those default setting imply, the authors mention that "We have carefully examined the different parameters and changed the information criteria for the stepwise selection procedure in GLM to 'Akaike Information Criteria (AIC)' and the number of terminal nodes in GBM to 6 as it is recommended by Friedman (2002)". I believe the reasoning behind the "default settings" has not been clarified yet.

We tuned the parameters of the algorithms and reran the models again, with no major changes in the modelled climatic suitability results. However, the AUC values were slightly improved and we decided to include the new modelling results in the manuscript. For a more detailed description, please refer to point 1.

About the Results section and Discussion section:In Supplementary file 4 there is not needed to show the background colours but simply the distribution of the data. The background does not really help to see the data.

Done as suggested. The background colours of Supplementary file 5 (former Supplementary file 4) have been removed.

Could you please explain why in the Global validation it was possible to estimate sensitivity but not specificity for T. rubrofasciata?

The sensitivity measures the percentage of true positives which means in this case congruence of projected and actual occurrence. The specificity measures the percentage of the correctly identified (true) negatives. However, since it is not possible to determine actual ‘climatic’ absences for a species that is currently spreading to non-endemic areas, the specificity cannot be calculated here.

In the Results section it is mentioned several times some agreements and disagreements between the model and the data for various species. For example, in the Discussion section "the models appear to slightly overestimate the potential distribution as it could be noted in the modelling of T. dimidiate". However, it is actually hard for the reader to note exactly where these potential overestimates are occurring. It will be great if you can have a figure (even if it is a set of figures in Supplementary file) where you show both the model predictions with the occurrence data on top so the reader can judge and understand where the model is fitting well and not that well, as you have done for T. rubrofasciata on Figure 2.

A set of figures was added (Supplementary file 4) depicting the modelling results as well as the occurrence of each species in South America obtained from the ‘Atlas of Chagas disease vectors in the Americas’ (Ceccarelli et al., 1998).

In Figure 1 (and also Figure 2) it is mentioned that "Hatched areas indicate regions where the projection is uncertain". There are two problems with this uncertainty:- The size of the panels makes the figures so small that it is impossible to actually see the hatched areas.- What does it mean "uncertain"? It should be clearly explained in the Materials and methods how such uncertainty was estimated. Is there a metric for such uncertainty?These problems with showing uncertainty in both Figure 1 and Figure 2 could be solved by having other similar figures exclusively for uncertainty.

A short description of the clamping mask (indicated as hatched areas in the figures) was given in the Materials and methods section. However, we have expanded on the explanation and tried to use a clearer vocabulary (Materials and methods section). Additionally, a figure has been added (Supplementary file 7) depicting the areas where the environmental variables are outside their trainings range.

To avoid confusion, I encourage authors to use a more cautious language when referring to climate suitability rather than actual presence of a particular species. For example in subsection “Potential distribution under current climate conditions” they mention "T. brasiliensis prefers dry and wet savannah climate as found in eastern Brazil and southern West Africa, northern and southern Central Africa and East Africa". But, in reality T. brasiliensis hasn't ever been found in Africa.

Done as suggested. We carefully examined the manuscript for inaccuracies and improved them if necessary (e.g. Results section).